# Mechanochemically Synthesized PEG-OTs as a Green Corrosion Inhibitor

**DOI:** 10.3390/polym17030422

**Published:** 2025-02-05

**Authors:** Qiannian Wang, Yuan Sang, Jiang Yang, Hailing Liu

**Affiliations:** College of Petrochemical Engineering, Liaoning Petrochemical University, Dandong Road West 1, Wanghua District, Fushun 113001, China

**Keywords:** mechanochemistry, ball mill, PEG, polymer corrosion inhibitor, solvent-free reaction, neat reaction, PEG modification, electrochemical impedance spectroscopy, hydrophobicity

## Abstract

Polymer corrosion inhibitors are reported to form dense films on carbon steel surfaces, and their thermostability enables survival in harsh downhole environments. In this paper, PEG-OTs was synthesized by mechanochemistry using ball mill by grafting tosyl on PEG. Using this solvent-free green chemistry, non-toxic PEG and PEG-OTs with various molecular weights (600, 2000, and 10,000 g/mol) were prepared and used as corrosion inhibitors. The corrosion inhibition performance of 5 × 10^−3^ mol/L inhibitors on Q235 carbon steel in 0.5 M HCl solution was investigated using static weight-loss, electrochemical impedance spectroscopy, polarization curves, SEM, and contact angle measurements. The results show that, after modification, PEG-OTs has an elevated corrosion inhibition effect compared to PEG. A maximum of 90% corrosion inhibition efficiency was achieved using static weight-loss. The morphology study shows that a dense film formed to protect carbon steel. Thanks to their polymeric structure, a higher molecular weight leads to better corrosion inhibition.

## 1. Introduction

Corrosion has been a major concern in carbon steel in the engineering industry, accounting for approximately 3–4% of the world’s GDP [1]. Within the oil and gas sector alone, corrosion-related costs are estimated at 170 billion USD, including about 463 million USD from downhole operations [2]. During petroleum production, mineral oxides are commonly removed by adding HCl to an oil-well [3,4]. This acid pickling process is a standard industrial cleaning method to eliminate mineral scale deposits [5,6]. However, adding HCl also introduces hydrogen ions that causes metallic oxidation. Once oxidized, the metal dissolves in solution and releases electrons. At the same time, an electric potential develops, forming an anodic site (where excess electrons accumulate) and a cathodic site (due to the reduction of hydrogen ions). In this electrochemical process, carbon steel is gradually dissolved [7,8,9]. To solve the corrosion problem, one of the most cost-effective approaches is to add corrosion inhibitors rather than investing in corrosion-resistant equipment or coating ordinary equipment with anti-corrosion materials [10,11,12,13]. Corrosion inhibitors include neutralizing inhibitors that decrease the concentration of hydrogen ions, scavenger that remove corrosive agents, and film forming corrosion inhibitors (commonly called corrosion inhibitors), which creates a barrier on the metal surface [14,15]. The barrier prevents metal dissolution by blocking anodic or cathodic reactions. Typically, these inhibitors contain polar groups to bind with metal surfaces and hydrophobic head groups that face away from the metal, forming an additional protection layer. The presence of π-electrons, heteroatoms, or aromatic rings can provide electron-donating sites that adsorb onto the metal surface [16,17,18,19]. However, many currently used corrosion inhibitors degrade at high temperatures in downhole environment. This problem can be solved by employing polymeric corrosion inhibitors, which generally exhibit superior thermal stability and can cover a larger metal surface area due to their repeating units. Finally, to align with sustainable chemistry principles, a greener synthesis route can be adopted to produce polymeric corrosion inhibitors [8].

Mechanochemistry provides a green approach to chemical synthesis by minimizing the use of solvents [20,21,22]. Currently, much of chemistry still relies heavily on solvents. For example, in the pharmaceutical industry, 85% of the chemicals used are solvents, and more than 20% of these solvents are lost during each recovery process [23], resulting in both economic loss and environmental pollution. Because most solvents are derived from petroleum, they can be hazardous, expensive, and energy-demanding. Therefore, solvent use must be minimized in organic synthesis [24,25]. Mechanochemistry addresses this need by using significantly fewer solvents [26]. Mechanochemistry refers to chemical reactions induced by the direct absorption of mechanical force [27,28]. The mechanical force can come from various types of mechanical actions, including impact, compression, shearing, stretching, and grinding [29,30]. Such actions create active sites for chemical reactivity, generating surfaces where particles can contact, coalesce, and react [26,31]. In particular, ball milling as a form of mechanochemistry is attracting increasing attention in the field of chemistry [32,33,34]. Recently, mechanochemistry has also been applied to modify terminal hydroxyl groups. For instance, Malca et al. demonstrated the mechanochemical modification of PEG by tosyl, bromine, thiol, carboxylic acid, and amine under solventless ball-milling conditions [35]. Moreover, Glassner et al. investigated the reaction of BCN-OH with tosyl chloride under ball-milling conditions to generate BCN-OTs [36].

PEG is recognized as a safe additive by the FDA and is used in cosmetics, toothpaste, drugs, and processed foods. The abundant presence of oxygen atoms allows PEG to resist corrosion [37,38,39]. Sorkhabi et al. used PEG directly to inhibit the carbon steel corrosion [40], while Ding et al. investigated aniline-tetramer-grafted PEG for corrosion inhibition [41]. Besides the heteroatom enabling PEG adsorption on the metal surface, its high thermal stability makes PEG a robust inhibitor for downhole environment. PEG begins to degrade at 350 °C [42]. In comparison, many common corrosion inhibitors degrade more easily, for example, morpholine degrades at 175 °C [43]. In this work, PEG was modified into PEG-OTs by ball milling, which is a straightforward green synthesis and avoids the use of solvents. The tosyl group brings both benzene and sulfur to PEG, providing extra affinity for the metal surface. The polymeric structure is also expected to exhibit better thermal stability than many common corrosion inhibitors. The corrosion inhibition performance of PEG and PEG-OTs on carbon steel in 0.5 M HCl solution was investigated through static corrosion tests, electrochemical impedance spectroscopy (EIS), Tafel plot curves, SEM, and contact angle measurements.

## 2. Materials and Methods

### 2.1. Materials

Polyethylene glycol (PEG_600_, PEG_2000_, PEG_10,000_), sodium hydroxide, potassium carbonate, and 4-toluenesulfonyl chloride were purchased from Aladdin Industrial Corporation (Los Angels, CA, USA) and were used directly without further purification. Anhydrous ethanol, acetone, HCl (38%), and methylene chloride were all purchased from Tianjin Damao Chemical Reagent Factory (Tianjin, China).

### 2.2. Ball Mill Machine

The planetary ball mill machine is the Premium 7 model from Fritsch manufacture in Burladingen, Germany, featuring a ZrO_2_ interior and 3 mm ZrO_2_ balls for grinding.

### 2.3. Synthesis

PEG_600_ (2 g, 1.67 mmol) and NaOH (320.6 mg, 4.008 mmol) were added to a ball mill vessel along with forty 3 mm zirconia balls. The ball mill was set to vibrate at 600 rpm for 30 min at an ambient temperature. Next, tosyl chloride (1.908 g, 5.02 mmol) and K_2_CO_3_ (923.2 mg, 3.34 mmol) were added to the vessel, and the ball milling was continued for 15 min. After the reaction, the product was dissolved in dichloromethane, yielding a white, insoluble suspension, which was then vacuum-filtered. The filtrate was concentrated via rotary evaporation at a 45 °C. The resulting product was transferred back into the ball mill vessel, followed by the addition of NaOH (1.908 g, 47.4 mmol). The mixture was then ball-milled for another 15 min to destroy any remaining excess tosyl chloride. Afterwards, the product was dissolved in dichloromethane and vacuum-filtered. The filtrate was extracted with saturated saline. Following extraction, the organic layer was dried over anhydrous magnesium sulfate for 30 min to remove any residual water. The dried product was then vacuum-filtered and concentrated by rotary evaporation. PEG_600_-OTs was obtained as a yellowish oil and was characterized by ^1^H NMR.

### 2.4. Static Corrosion Tests

The static corrosion behavior of Q235 carbon steels was tested using weight-loss experiments. Specimen for weight-loss measurements measured 40 mm × 13 mm × 2 mm (*A*). Before each test, each specimen was cleaned to remove oil residuals from the new carbon steel. First, it was soaked in acetone and ultrasonically cleaned for 5 min. Next, it was immersed in absolute ethanol and ultrasonically cleaned for 5 min. Then, it was taken out, blow-dried, and stored in a desiccator. The specimen was weighed (*m*_1_) on an electronic balance with an accuracy of 0.1 mg. At this point, the specimen was ready for the static corrosion test. The corrosive solution was a 0.5 M HCl solution in double-distilled water containing 5 × 10^−3^ mol/L inhibitors. The static corrosion test was conducted without stirring at room temperature for 24 h (*t*). To remove corrosion products and measure the weight of undissolved carbon steel, the specimens were cleaned again. First, each specimen was immersed in Clarke’s solution and ultrasonically cleaned. Clarke’s solution contains 6 g of hexamethylenetetramine (C_6_H_12_N_4_), 500 mL of deionized water, and 500 mL of 38% HCl. Then, the specimen was placed in 10 wt% NaOH solution and ultrasonically cleaned. Afterwards, it was rinsed with ethanol, dried with a heat gun, and weighed again (*m*_2_). Finally, the corrosion rate and corrosion inhibition rate (*ν*) were calculated as described below(1)$$\nu_{i}=\frac{10^{6}\times \Delta m_{i}}{A_{i}\times \Delta t}$$
where *ν* denotes the corrosion rate in g·m^−2^·h^−1^; *i* indicates each specimen; Δ*m_i_* represents the mass difference before and after corrosion (in grams), where Δ*m_i_* = *m*_2_ − *m*_1_; *A_i_* is the surface area of the specimen (in mm^2^); and Δ*t* is the corrosion reaction time (in hours).

Here, ν_0_ is the corrosion rate of the blank specimen with no inhibitor. The corrosion inhibition rate (*η*) of the steel sheet in the static weight-loss test can be calculated according to Equation (2) below:(2)$$\eta =\frac{\nu_{0}-\nu }{\nu_{0}}\times 100\%$$

Three tests were conducted for each sample, and the average values were used.

### 2.5. Electrochemical Tests

All atmospheric electrochemical measurements were performed using a three-electrode system in a five-port corrosion test cell, with Q235 carbon steel (working area 1 cm^2^) as the working electrode. The Potentiostat is Gamry instruments manufacture and mode 39016 from Philadelphia, PA, USA. A platinum sheet was used as the counter electrode. A saturated calomel electrode was used as the reference electrode. To eliminate the ohmic drop, the saturated calomel electrode was immersed in a salt bridge and connected to the electrolyte via a Luquin capillary. Before all electrochemical tests, an open-circuit potential (OCP) test was carried out for 1 h to ensure that the electrode potential deviation was less than 5 mV, indicating a stable state. The electrochemical impedance spectroscopy (EIS) measurements were performed over a frequency range of 10^5^~10^−2^ Hz with an AC amplitude of 10 mV, and the resulting data were fitted using ZSimpWin software version 3.50. The scanning range of the potentiodynamic polarization test was from −200 mV (vs. OCP) to +200 mV (vs. OCP) at a scanning rate of 0.33 mV·s^−1^. All linear polarization (LPR) tests were conducted from −10 mV (vs. OCP) to +10 mV (vs. OCP) at a scanning rate of 0.167 mV·s^−1^.

### 2.6. SEM Analysis

A Hitachi SU801 scanning electron microscope from Tokyo, Japan and a Bruker Xflash 5030 detector from Berlin, Germany were used to observe the morphology of the samples. Specimens measuring 10 mm × 10 mm × 2 mm were first cleaned to remove residual oil from the new Q235 carbon steel. Each specimen was soaked in acetone and ultrasonically cleaned for 5 min. Next, it was immersed in absolute ethanol for ultrasonically cleaned for another 5 min. Then, it was blow-dried, placed in a desiccator, and dried for 20 min. The reference specimen was ready for testing, while the test specimens were immersed in 0.5 M HCl solution for 24 h without and with 5 × 10^−3^ mol/L PEG_10,000_ and PEG_10,000_-OTs inhibitors. To remove the corrosion products, the test specimens were immersed in Clarke’s solution (6 g of hexamethylenetetramine (C_6_H_12_N_4_) + 500 mL of deionized water + 500 mL of 38% HCl), followed by 5 min of ultrasonic cleaning to remove any remaining corrosion products. The specimens were then rinsed with ethanol, dried with a heat gun, and prepared for SEM analysis.

### 2.7. Contact Angle Measurements

The water contact angle (*θ*) on the carbon steel surface was measured on a Dataphysics OCA 15EC using the solid-drop method. Before measurement, the carbon steel specimens were polished with 100, 400, 800, 1000, and 1500 grit sandpapers. All contact angles were measured with the 2 μL liquid droplets containing 0.5 M HCl with or without 5 × 10^−3^ mol/L PEG_10,000_ and PEG_10,000_-OTs inhibitors. The contact angles were measured once the droplet stabilized on the specimen surface (within 1–2 min). Each reported value is the average of the five measurements, ensuring that the standard deviation is less than 3°.

## 3. Results and Discussions

### 3.1. Synthesis of PEG-OTs

The synthesis of PEG-OTs was carried out as shown in Figure 1, following the procedure described in reference [35]. First, PEG and NaOH were ball-milled at 600 rpm with forty 3 mm-diameter ZrO_2_ balls for 30 min. K_2_CO_3_ and Tosyl chloride were then added and ball-milled for another 15 min. After the reaction and dissolution, additional NaOH was introduced and ball-milled again to remove any excess Tosyl chloride. This reaction is notable because it uses a neat condition without solvent, enabled by ball-milled mechanochemistry. Using this method, three samples of PEG_600_-OTs, PEG_2000_-OTs, and PEG_10,000_-OTs were synthesized and characterized by ^1^H NMR and FT-IR. In the ^1^H NMR in Figure 1, the proton “d” shifted from 1.88 ppm to 4.09 ppm after modification. Moreover, the proton “a” at 3.02 ppm, proton “b” at 7.27 ppm, and “c” at 7.72 appeared. The integrals match well and confirm the successful synthesis. The FT-IR of PEG_600_ and PEG_600_-OTs are presented in Figure 2. In the spectrum of PEG_600_, the absorption peaks at 3427 cm^−1^ and 2876 cm^−1^ correspond to the stretching vibrations of -OH and -CH_2_-, respectively. The absorption peak at 1116 cm^−1^ corresponds to the anti-symmetric stretching vibration of C-O-C. After modification, the FT-IR spectrum of PEG_600_-OTs indicates that the main chain structure of PEG_600_ remains unchanged. The decreased absorption peak at 3427 cm^−1^ shows that the terminal hydroxyl of PEG_600_ were consumed. Additionally, the S=O stretching band at 1719 cm^−1^ appears in PEG_600_-OTs.

### 3.2. Static Weight-Loss

All samples including PEG_600_, PEG_600_-OTs, PEG_2000_, PEG_2000_-OTs, PEG_10,000_, and PEG_10,000_-OTs were evaluated by a static weight-loss method. The weight-loss test was carried out at 5 × 10^−3^ mol/L inhibitors concentration in 0.5 M HCl solution on Q235 carbon steel for 24 h. The carbon steel was cleaned after corrosion according to reference [44]. The mass of carbon steel before and after testing was compared, and the corrosion rate was calculated according to Equation (1). Then, the corrosion inhibition rate was calculated according to Equation (2) and is shown in Figure 2. It is evident that, after tosyl modification, PEG demonstrates better corrosion inhibition, indicating that the tosyl group contributes to the inhibition effect. Moreover, as the molecular weight increases, the PEG/PEG-OTs exhibit improved corrosion inhibition. This can be explained by the fact that, at the same molar concentration, the number of tosyl groups is the across different molecular weights, while a higher molecular weight provides more repeating -CH_2_CH_2_O- units, thus enhancing corrosion inhibition.

### 3.3. Electrochemical Results

Electrochemical Impedance Spectroscopy (EIS) was used to further evaluate the corrosion inhibition performance. The carbon steel was immersed in 0.5 M HCl solution for 24 h at room temperature and then tested. The impedance spectra exhibit a single semicircle, indicating charge transfer between the electrode and the solution. The transfer process is relevant to the corrosion reaction rates of carbon steels in the HCl [45]. As shown in Figure 3, the Nyquist plots reveal a capacitive arc. The slope of the low-frequency line corresponds to the Warburg coefficient, which measures the mass-transfer rate. A high slope indicates that mass transfer is a limiting factor in this system. After tosyl modification, the arc radius increases, indicating improved corrosion inhibition in Figure 3a–c. Furthermore, as the molecular weight increases, the arc radius also increases Figure 3d,e, suggesting an even better corrosion inhibition. Bode plots are presented in Appendix A of the Appendix A, along with the equivalent circuit diagram in Appendix A, and the EIS results in Table 1. These results are consistent with the static weight-loss results.

As shown in Table 1, the equivalent circuit model was used to fit the AC impedance data by ZSimpWin Software. Here, *R*_ct_, the charge transfer resistance is the key. Higher *R*_ct_ means the corrosion has less possibility in contact with the carbon steel, and the corrosion is inhibited.

### 3.4. Tafel Polarization Results

Dynamic potential polarization tests were conducted on the inhibitors to obtain the corrosion potential (*E*_corr_) and corrosion current density (*I*_corr_), as shown in Table 2. *E*_corr_ represents the potential of a specific corrosion system when it reaches a steady state without an applied external voltage, reflecting the material’s tendency to corrode. As *E*_corr_ increases, the steel’s corrosion susceptibility decreases. Tafel curves, obtained by scanning the potential of the working electrodes (PEG and PEG-OTs) in 0.5 M HCl, are presented in Figure 4. After the addition of corrosion inhibitors, the Tafel polarizations shift to the left, and the *I*_corr_ values of steel were reduced, indicating that corrosion was inhibited. As shown in Figure 4a–c, the corrosion resistance efficiency of PEG-OTs is higher than that of PEG. Meanwhile, Figure 4d,e illustrate that the corrosion inhibition efficiency of both PEG and PEG-OTs increases with increasing molecular weight of the corrosion inhibitor, consistent with the impedance test results.

Table 2 shows that the charge transfer resistance *R*_ct_ is lowest for the blank sample at 51.6 Ω·cm^2^, with a film resistance *R*_f_ of 4.8 Ω·cm^2^. The resistivities of other samples, from highest to lowest, are 262.3 Ω·cm^2^ (PEG_10,000_-OTs) > 238.5 Ω·cm^2^ (PEG_10,000_) > 189.2 Ω·cm^2^ (PEG_2000_-OTs) > 175.2 Ω·cm^2^ (PEG_2000_) > 125.6 Ω·cm^2^ (PEG_600_-OTs) > 89.5 Ω·cm^2^ (PEG_600_). The sample with PEG_10,000_-OTs shows the highest impedance value, indicating that it has excellent corrosion resistance. The results show that the self-etching current density of polyethylene glycol PEG_10,000_-OTs is the lowest, only 60.663 μA·cm^−2^ in Table 2. The results show that PEG_10,000_-OTs have the best corrosion inhibition effect on Q235 steel. It is found that the corrosion resistance of the inhibitor decreases with the increase in molecular weight, and its corrosion inhibition rate is lower than that of the traditional inhibitor. This conclusion is consistent with the AC impedance measurement results.

### 3.5. Morphological Characteristics

The surface morphologies of Q235 carbon steel samples under different conditions—before corrosion, after corrosion without inhibitor, after corrosion with 5 × 10^−3^ mol/L of PEG_10,000_, after corrosion with 5 × 10^−3^ mol/L of PEG_10,000_-OTs were investigated by SEM, and the results are shown in Figure 5. Prior to corrosion, the carbon steel surface in Figure 5a appears generally smooth, with only slight polishing marks. The steel specimens were immersed in 0.5 M HCl solution after 24 h and then cleaned for SEM observation. Without inhibitor, the surface shows severe corrosion damage with deep pits in Figure 5b, indicating significant material loss. By contrast, specimens treated with PEG_10,000_ and PEG_10,000_-OTs exhibit fewer pits due to a protective film forming on the surface, visible as the lighter-colored layer in Figure 5c,d. Notably, the sample with PEG_10,000_-OTs shows a smoother surface, indicating enhanced corrosion protection. This phenomenon is also evident in the uncleaned carbon steel picture in Appendix A. After immersed in 0.5 M HCl for 24 h, the specimens are taken out immediately for a straightforward picture. As shown in Appendix A, the carbon steel is corroded to a dark color with no inhibitor. With the inhibitor of PEG_10,000_, the specimen is slightly corroded and mostly light colored in Appendix A. The specimen with PEG_10,000_-OTs is barely corroded in Appendix A. Compared to the polished carbon steel, the specimen was covered by a dense film outside the metallic luster. Consequently, the morphological analysis aligns with the electrochemical measurements, confirming that the PEG_10,000_-OTs creates a superior protective film on the carbon steel surface and thereby improves corrosion resistance.

### 3.6. Contact Angle

To evaluate hydrophobicity, contact angle measurements were performed. As shown in Figure 6, cleaned carbon steel samples were placed at the bottom, and droplets of 0.5 M HCl solution (with and without 5 × 10^−3^ mol/L PEG_10,000_ and PEG_10,000_-OTs) were applied. Five measurements were taken at different spots on each sample, and the average value was recorded, ensuring a standard deviation of less than 3°. The contact angle of the blank solution was 57° in Figure 6a. In the presence of PEG_10,000_ inhibitor, the angle increased to 63.5° in Figure 6b. With PEG_10,000_-OTs inhibitor, the angle increased further to 81.1° in Figure 6c. The larger contact angle indicates greater hydrophobicity, helping to protect the steel from the HCl solution. The increase in contact angle also confirms the adsorption of inhibitors onto the carbon steel surface. Moreover, PEG_10,000_-OTs provides better inhibition than PEG_10,000_. Thus, the tosyl functional group improves corrosion inhibition.

## 4. Conclusions

In this paper, PEG was modified to PEG-OTs by mechanochemistry, a solvent-free process. Using this green synthesis, three PEG-OTs with different molecular weights (600, 2000, 10,000 g/mol) were prepared. Through static corrosion test, EIS, and Tafel polarization measurements, we found that PEG-OTs showed elevated corrosion inhibition on carbon steel in the presence of 0.5 M HCl compared with PEG. Moreover, at the same concentration, higher molecular weight PEG exhibited enhanced corrosion inhibition. The inhibition correlates with increased hydrophobicity, as indicated by contact angle measurements. SEM images confirm the formation of a protective film responsible for corrosion inhibition. Therefore, a polymeric PEG-OTs corrosion inhibitor was successfully synthesized using green chemistry.

## Data Availability

All data generated or analyzed during this study are included in this published article.

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
