# Peer review of "Mechanochemically Synthesized PEG-OTs as a Green Corrosion Inhibitor"

_polymers, 2025, doi:10.3390/polym17030422_

Round 1

Reviewer 1 Report

Comments and Suggestions for Authors

Manuscript Title: Mechanochemistry Synthesized PEG-OTs as a Polymeric Green Corrosion Inhibitor for Carbon Steel

Summary

The manuscript explores the synthesis of PEG-OTs using mechanochemical methods and evaluates its corrosion inhibition performance for carbon steel in acidic environments. The work presents an innovative approach combining green chemistry principles with corrosion protection strategies. However, the manuscript requires significant revisions in several areas to improve clarity, scientific rigor, and overall quality.

Page 1:

Title: Consider rephrasing the title for conciseness and impact, e.g., “Mechanochemically Synthesized PEG-OTs as a Green Corrosion Inhibitor.”

Abstract:

The abstract lacks quantitative data on corrosion inhibition efficiency. Including key numerical results (e.g., inhibition efficiency) would strengthen its impact. Other results such as isotherm model and inhibitor type (andic, cathodic or mixed) should be added.

The full name of PEG-OTs should be written when it is mentioned for the first time.

Grammatical error: "PEG-OTs was synthesized" should be "PEG-OTs were synthesized."

Keywords: While appropriate, consider including terms like "hydrophobicity" and "electrochemical impedance spectroscopy" to enhance discoverability.

Page 2:

Introduction:

The context provided on corrosion costs and mechanisms is well-framed but overly lengthy. Streamline the discussion to maintain focus on the significance of polymeric corrosion inhibitors.

Grammatical issues:

"This problem can be solved by employing polymeric corrosion inhibitors" should be rephrased to specify the advantages of PEG-OTs.

The introduction should briefly outline the novelty of the mechanochemical synthesis used.

Page 3:

Mechanochemistry:

The description of mechanochemistry is informative but lacks recent references. Update the citations to include works from 2023-2024.

Improve clarity in the statement: "chemical reactions induced by direct absorption of mechanical force."

Grammar:

"minimizing the use of solvents" could be replaced with "reducing solvent usage" for conciseness.

Tense inconsistency is observed throughout the section.

Page 4:

Experimental Section:

Synthesis procedure of PEG-OTs should be supported by references.

Why did authors use 0.5 M HCl solution?

Why didn’t the authors utilize different concentrations of inhibitors in the weight loss tests?

The brand and model of the potentiostat should be specified.

Results and discussion:

Please modify the sub-title to characterization of PEG-OTs instead of synthesis of PEG-OTs.

Grammatical issues:

"was vacuum-filtered and concentrated by rotary evaporation" should be "vacuum-filtered, then concentrated using rotary evaporation."

Page 5:

Figures and Spectral Analysis:

Figures 1 and 2 lack adequate labelling. Add clear axis titles, units, and legends for better interpretation.

The FT-IR analysis discussion is insufficient. Explicitly link observed spectral changes to structural modifications in PEG-OTs.

Page 7:

Electrochemical Impedance Spectroscopy (EIS):

The connection between Nyquist plot arc radii and corrosion inhibition performance is not clearly explained. Elaborate on the underlying mechanisms.

Grammatical issue: "suggestion even better corrosion inhibition" should be "indicating enhanced corrosion inhibition." Page 8:

Tafel Polarization Results:

Data presentation is robust but over-reliant on supplementary materials. Key findings, such as corrosion potential changes, should be highlighted in the main text.

Ensure consistent capitalization of "Tafel" throughout.

Page 10:

Contact Angle Measurements:

This section successfully connects hydrophobicity to corrosion resistance. However, clarify whether the contact angles reported are equilibrium values.

Grammatical issue: "Thus, the tosyl functional group improves corrosion inhibition" should expand on the role of hydrophobicity and adsorption.

Page 11:

Conclusions:

The conclusions are concise but lack specific data points. Include numerical results for corrosion inhibition efficiency to strengthen the summary.

Discuss limitations of the study, such as scalability or performance in varied conditions.

References:

Many references are outdated. Replace some with recent works (2020-2024) to enhance relevance.

Ensure consistency in formatting.

General Comments

Strengths:

The manuscript addresses an important challenge in corrosion inhibition with a new mechanochemical approach.

Experimental results are comprehensive, covering a wide range of characterization techniques (e.g., EIS, Tafel plots, LPR, SEM).

Weaknesses:

The writing quality needs improvement, with frequent grammatical errors and inconsistent tenses.

The discussion of results lacks sufficient depth in explaining underlying mechanisms.

Figures and tables require better labelling and integration into the main text.

A single concentration was used for each inhibitor. 

The isotherm models were not investigated.

The temperature effect was not examined.

Suggestions for Improvement:

Writing Quality: Revise the manuscript for grammatical accuracy, clarity, and conciseness.

Different concentrations: using of various concentrations of inhibitors is highly recommended.

The inhibition efficiency should be examined at different temperature to investigate the thermal stability of the inhibitors.

Figures: Improve figure captions and ensure all axes, units, and legends are clear.

Discussion: Expand on mechanisms behind PEG-OTs performance, particularly in relation to hydrophobicity and molecular weight effects.

References: Update the bibliography to include recent studies.

Comments on the Quality of English Language

minor modifications are required (detailed suggestions are shown in my comments).

Author Response

Summary

The manuscript explores the synthesis of PEG-OTs using mechanochemical methods and evaluates its corrosion inhibition performance for carbon steel in acidic environments. The work presents an innovative approach combining green chemistry principles with corrosion protection strategies. However, the manuscript requires significant revisions in several areas to improve clarity, scientific rigor, and overall quality.

Page 1:

Title: Consider rephrasing the title for conciseness and impact, e.g., “Mechanochemically Synthesized PEG-OTs as a Green Corrosion Inhibitor.”

According to the reviewer’s suggestion, the title is changed.

Abstract:

The abstract lacks quantitative data on corrosion inhibition efficiency. Including key numerical results (e.g., inhibition efficiency) would strengthen its impact. Other results such as isotherm model and inhibitor type (andic, cathodic or mixed) should be added.

According to the reviewer’s suggestion, the key numerical results and inhibition mechanism is added. “A maximum of 90% corrosion inhibition efficiency was achieved using static weight-loss. The morphology study shows that a dense film formed to protect carbon steel.”

Here we focus on the polymer synthesis as well as the protection behavior. The film on the carbon steel is observed as the mechanism reveal.

The full name of PEG-OTs should be written when it is mentioned for the first time.

According to the reviewer’s suggestion, this is explained in the abstract “In this paper, PEG-OTs was synthesized by mechanochemistry using ball mill by grafting tosyl on PEG”

Here, OTs is the common writing of tosyl group in organic chemistry, and there is no full name of it.

Grammatical error: "PEG-OTs was synthesized" should be "PEG-OTs were synthesized."

OTs is the common writing of tosyl group in organic chemistry. This is singular form instead of plural form.

Keywords: While appropriate, consider including terms like "hydrophobicity" and "electrochemical impedance spectroscopy" to enhance discoverability.

According to the reviewer’s suggestion, the keywords are added.

Page 2:

Introduction:

The context provided on corrosion costs and mechanisms is well-framed but overly lengthy. Streamline the discussion to maintain focus on the significance of polymeric corrosion inhibitors.

As this manuscript is submitting to journal “Polymers”, the authors believe a significant part of the readers is from polymer field, especially from polymer synthesis. These readers may not be familiar with the background of corrosion concept. To address the significance of corrosion inhibitors, the background needs to be comprehensive. Polymeric corrosion inhibitors surely rise an interesting topic as more repeating units could cover carbon steel surface. We highly recommend a review for additional information about it.

Tiu, Brylee David B., and Rigoberto C. Advincula. "Polymeric corrosion inhibitors for the oil and gas industry: Design principles and mechanism." Reactive and Functional Polymers 95 (2015): 25-45.

Grammatical issues:

"This problem can be solved by employing polymeric corrosion inhibitors" should be rephrased to specify the advantages of PEG-OTs.

The advantages of PEG-OTs have already been shown as heteroatom in the introduction.

The introduction should briefly outline the novelty of the mechanochemical synthesis used.

The second paragraph in the introduction tells the advantages of mechanochemical synthesis. Briefly, different from conventional chemical reaction in a flask and solvent or water with stir, mechanical synthesis happens in milled condition and no solvent is used. Organic solvents can be toxic and pricy. Organic synthesis takes lots of solvents. The avoidance of solvent of mechanical synthesis leads to green chemistry. So, this is the significance or novelty of this synthesis.

Page 3:

Mechanochemistry:

The description of mechanochemistry is informative but lacks recent references. Update the citations to include works from 2023-2024.

The authors include references useful to our research and there ARE papers published recently cited as below,

Wang, X.; Yang, J.; Chen, X. 2-Benzylsulfanyl-1H-benzimidazole and its mixture as highly efficient corrosion inhibitors for carbon steel under dynamic supercritical CO2 flow conditions. Corros. Sci. 2024, 235, 112170-112190.

Zhang, W.; Wang, S.; Guo, Z.; Luo, J.; Zhang, C.; Zhang, G. Heterocyclic group end-capped polyethylene glycol: Synthesis and used as corrosion inhibitors for mild steel in HCl medium. J. Mol. Liq. 2023, 388.

Jafter, O.F.; Lee, S.; Park, J.; Cabanetos, C.; Lungerich, D. Navigating ball mill specifications for theory-to-practice reproduc-ibility in mechanochemistry. Angew Chem Int Ed Engl. 2024, 63, e202409731.

Guan, X.; Sang, Y.; Liu, H. Ball-milled click chemistry: a solvent-free green chemistry. Progr. Chem. 2024, 36, 401-415.

Sang, Y.; Guan, X.; Liu, H.; Yang, J.; Li, S.; Wang, Q.; Huang, X.; Wang, Y.; Yu, Q.; Li, M. Solvent-free lignin modification through ball-milled CuAAc. ACS Sustain. Chem. Eng. 2024, 12, 13700-13710.

Ahmed, M.A.; Amin, S.; Mohamed, A.A. Current and emerging trends of inorganic, organic and eco-friendly corrosion in-hibitors. RSC Adv. 2024, 14, 31877-31920.

Dong, L.; Ma, Y.; Jin, X.; Feng, L.; Zhu, H.; Hu, Z.; Ma, X. High-efficiency corrosion inhibitor of biomass-derived high-yield carbon quantum dots for Q235 carbon steel in 1 M HCl solution. ACS Omega. 2023, 8, 46934-46945.

Jalab, R.; Ali, A.B.; Khaled, M.; Abouseada, M.; AlKhalil, S.; Al-Suwaidi, A.; Hamze, S.; Hussein, I. Novel polyepoxysuccinic acid-grafted polyacrylamide as a green corrosion inhibitor for carbon steel in acidic solution. ACS Omega. 2023, 8, 16673-16686.

Improve clarity in the statement: "chemical reactions induced by direct absorption of mechanical force."

This is a truth and is not a mistake saying. It can be found in many references. For instance,

Shen, Feng, et al. "Recent advances in mechanochemical production of chemicals and carbon materials from sustainable biomass resources." Renewable and sustainable energy reviews 130 (2020): 109944.

This paper has been cited by 179 times and the authors believe this saying is not wrong.

Grammar:

"minimizing the use of solvents" could be replaced with "reducing solvent usage" for conciseness. Tense inconsistency is observed throughout the section.

The authors believe “minimizing” and “reduce” has different meaning. Minimizing means use very small amount of solvent or zero solvent, which is very suitable to mechanochemistry. Reduce mean less solvent, which is not correct for mechanochemistry.

Page 4:

Experimental Section:

Synthesis procedure of PEG-OTs should be supported by references.

If the reviewer could read more carefully, the reference 44 is cited in the result and discussion part.

Why did authors use 0.5 M HCl solution?

Why didn’t the authors utilize different concentrations of inhibitors in the weight loss tests?

These two questions can be combined and answered together. The acid condition is chosen according to reference from ACS Omega as cited in manuscript. Of course different concentrations of HCl can be done. But in this pioneer work, our emphasis is also on the mechanochemical polymeric modification. We can do tons of characterization but we need to find the focus and make sure our story has a center point, that is a green polymer on corrosion inhibition, rather than a new corrosion inhibitor found and tons of tests done to make sure it works in every acid condition.

The brand and model of the potentiostat should be specified.

The test is done by flask and Gamry instrument 39016, this isadded in the manuscript

Results and discussion:

Please modify the sub-title to characterization of PEG-OTs instead of synthesis of PEG-OTs.

The characterization is used to proof the success synthesis. Well, the importance of this part is synthesis.

Grammatical issues:

"was vacuum-filtered and concentrated by rotary evaporation" should be "vacuum-filtered, then concentrated using rotary evaporation."

The original paragraph says “Afterwards, the product was dissolved in dichloromethane and vacuum-filtered. The filtrate was extracted with saturated saline. Following extraction, the organic layer was dried over anhydrous magnesium sulfate for 30 min to remove any residual water. The dried product was then vacuum-filtered, and concentrated by rotary evaporation. PEG600-OTs was obtained as a yellowish oil and was characterized by 1H NMR.” The authors believe this saying could connect the sentences before and after.

Page 5:

Figures and Spectral Analysis:

Figures 1 and 2 lack adequate labelling. Add clear axis titles, units, and legends for better interpretation.

If the reviewer gets to know NMR and FTIR, the y-axis are surely fine as shown in the manuscript.

The FT-IR analysis discussion is insufficient. Explicitly link observed spectral changes to structural modifications in PEG-OTs.

“The decreased absorption peak at 3427 cm−1 shows that the terminal hydroxyl of PEG600 were consumed. Additionally, the S=O stretching band at 1719 cm-1 appears in PEG600-OTs.” This is discussion about FT-IR and already shown.

Page 7:

Electrochemical Impedance Spectroscopy (EIS):

The connection between Nyquist plot arc radii and corrosion inhibition performance is not clearly explained. Elaborate on the underlying mechanisms.

The authors stated the bigger radii means better inhibition. The authors emphasis on synthesis and some application of polymer instead of comprehensive corrosion inhibition study.

Grammatical issue: "suggestion even better corrosion inhibition" should be "indicating enhanced corrosion inhibition." Page 8:

The authors don’t see a problem here.

Tafel Polarization Results:

Data presentation is robust but over-reliant on supplementary materials. Key findings, such as corrosion potential changes, should be highlighted in the main text.

According to the reviewer’s suggestion, the table S1 and S2 are moved to manuscript.

Ensure consistent capitalization of "Tafel" throughout.

According to the reviewer’s suggestion, it is capitalized.

Page 10:

Contact Angle Measurements:

This section successfully connects hydrophobicity to corrosion resistance. However, clarify whether the contact angles reported are equilibrium values.

Yes, it is

Grammatical issue: "Thus, the tosyl functional group improves corrosion inhibition" should expand on the role of hydrophobicity and adsorption.

Yes this is stated in the 3.5

Page 11:

Conclusions:

The conclusions are concise but lack specific data points. Include numerical results for corrosion inhibition efficiency to strengthen the summary.

This manuscript implies a polymeric corrosion inhibitor. However, the inhibition effect is not as high as commercial inhibitor. So, the numerical number is not stated here.

Discuss limitations of the study, such as scalability or performance in varied conditions.

Limitation is obviously, corrosion inhibition is not as high, or else it will be a break through paper in “Corrosion Science”.

References:

Many references are outdated. Replace some with recent works (2020-2024) to enhance relevance.

The authors include references useful to our research and there ARE papers published recently cited as below,

Wang, X.; Yang, J.; Chen, X. 2-Benzylsulfanyl-1H-benzimidazole and its mixture as highly efficient corrosion inhibitors for carbon steel under dynamic supercritical CO2 flow conditions. Corros. Sci. 2024, 235, 112170-112190.

Zhang, W.; Wang, S.; Guo, Z.; Luo, J.; Zhang, C.; Zhang, G. Heterocyclic group end-capped polyethylene glycol: Synthesis and used as corrosion inhibitors for mild steel in HCl medium. J. Mol. Liq. 2023, 388.

Jafter, O.F.; Lee, S.; Park, J.; Cabanetos, C.; Lungerich, D. Navigating ball mill specifications for theory-to-practice reproduc-ibility in mechanochemistry. Angew Chem Int Ed Engl. 2024, 63, e202409731.

Guan, X.; Sang, Y.; Liu, H. Ball-milled click chemistry: a solvent-free green chemistry. Progr. Chem. 2024, 36, 401-415.

Sang, Y.; Guan, X.; Liu, H.; Yang, J.; Li, S.; Wang, Q.; Huang, X.; Wang, Y.; Yu, Q.; Li, M. Solvent-free lignin modification through ball-milled CuAAc. ACS Sustain. Chem. Eng. 2024, 12, 13700-13710.

Ahmed, M.A.; Amin, S.; Mohamed, A.A. Current and emerging trends of inorganic, organic and eco-friendly corrosion in-hibitors. RSC Adv. 2024, 14, 31877-31920.

Dong, L.; Ma, Y.; Jin, X.; Feng, L.; Zhu, H.; Hu, Z.; Ma, X. High-efficiency corrosion inhibitor of biomass-derived high-yield carbon quantum dots for Q235 carbon steel in 1 M HCl solution. ACS Omega. 2023, 8, 46934-46945.

Jalab, R.; Ali, A.B.; Khaled, M.; Abouseada, M.; AlKhalil, S.; Al-Suwaidi, A.; Hamze, S.; Hussein, I. Novel polyepoxysuccinic acid-grafted polyacrylamide as a green corrosion inhibitor for carbon steel in acidic solution. ACS Omega. 2023, 8, 16673-16686.

Ensure consistency in formatting.

General Comments

Strengths:

The manuscript addresses an important challenge in corrosion inhibition with a new mechanochemical approach.

Experimental results are comprehensive, covering a wide range of characterization techniques (e.g., EIS, Tafel plots, LPR, SEM).

Weaknesses:

The writing quality needs improvement, with frequent grammatical errors and inconsistent tenses.

The discussion of results lacks sufficient depth in explaining underlying mechanisms.

Figures and tables require better labelling and integration into the main text.

A single concentration was used for each inhibitor.

The isotherm models were not investigated.

The temperature effect was not examined.

Suggestions for Improvement:

Writing Quality: Revise the manuscript for grammatical accuracy, clarity, and conciseness.

Different concentrations: using of various concentrations of inhibitors is highly recommended.

The inhibition efficiency should be examined at different temperature to investigate the thermal stability of the inhibitors.

Figures: Improve figure captions and ensure all axes, units, and legends are clear.

Discussion: Expand on mechanisms behind PEG-OTs performance, particularly in relation to hydrophobicity and molecular weight effects.

References: Update the bibliography to include recent studies.

The authors appreciate the reviewer’s feedback for so many words typing. We would appreciate much more if a mentor could write this feedback instead of a graduate student.

Reviewer 2 Report

Comments and Suggestions for Authors

In my opinion, the manuscript polymers-3455784 entitled " Mechanochemistry Synthesized PEG-OTs as a Polymeric Green Corrosion Inhibitor for Carbon Steel" is suitable for publication in the journal Polymers after minor correction.

The manuscript includes important issues that may interest a broad material engineering audience. Applied research techniques are adequate for the studies taken. The language is understandable. Conclusions are related to the obtained test results.

The manuscript contains important issues that may interest a wide audience of materials engineering readers. Several research techniques appropriate to the research undertaken were used. The manuscript's language is understandable, with clear descriptions of both the synthesis of inhibitors and the electrochemical tests and surface morphology studies. The conclusions refer to the test results obtained. I believe Tables S1 and S2, included in the supplementary materials, should be included in the manuscript. The main text lacks the results of polymer efficiency, and the data obtained in EIS and LSV studies are more readable in the form of a table. I suggest shortening the part describing the research techniques and including the Tables in the main text.

Author Response

In my opinion, the manuscript polymers-3455784 entitled " Mechanochemistry Synthesized PEG-OTs as a Polymeric Green Corrosion Inhibitor for Carbon Steel" is suitable for publication in the journal Polymers after minor correction.

The manuscript includes important issues that may interest a broad material engineering audience. Applied research techniques are adequate for the studies taken. The language is understandable. Conclusions are related to the obtained test results.

The manuscript contains important issues that may interest a wide audience of materials engineering readers. Several research techniques appropriate to the research undertaken were used. The manuscript's language is understandable, with clear descriptions of both the synthesis of inhibitors and the electrochemical tests and surface morphology studies. The conclusions refer to the test results obtained. I believe Tables S1 and S2, included in the supplementary materials, should be included in the manuscript. The main text lacks the results of polymer efficiency, and the data obtained in EIS and LSV studies are more readable in the form of a table. I suggest shortening the part describing the research techniques and including the Tables in the main text.

According to the reviewer’s suggestion, the table S1 and table S2 are moved to the manuscript from SI. The research techniques are accordingly shortened.

Round 2

Reviewer 1 Report

Comments and Suggestions for Authors

The authors effectively addressed all the points raised in the Round 1 report.